

# Beliefs are multidimensional and vary in stability over time - psychometric properties of the Beliefs and Values Inventory (BVI)

Joseph M. Barnby[1], Vaughan Bell[1,3], Luke Sheridan Rains[4], Mitul A. Mehta[1,*] and Quinton Deeley[2,*]

[1] Cultural and Social Neuroscience Research Group, Department of Neuroimaging, Kings College London, Institute of Psychiatry, Psychology, and Neuroscience, London, United Kingdom
[2] Cultural and Social Neuroscience Research Group, Forensic and Neurodevelopmental Sciences, Kings College London, Institute of Psychiatry, Psychology, and Neuroscience, London, United Kingdom
[3] Research Department of Clinical, Educational, and Health Psychology, University College London, London, United Kingdom
[4] Division of Psychiatry, University College London, London, United Kingdom
[*] These authors contributed equally to this work.

Corresponding author
Joseph M. Barnby,
joe.barnby@kcl.ac.uk

## ABSTRACT

The cognitive processes underlying belief are still obscure. Understanding these processes may lead to more targeted treatment to better address functional impairment, such as occurs with delusions. One way in which this might be accomplished is to understand healthy, everyday beliefs, and how these may relate to characteristics observed in delusions. As yet, no such measure exists to accurately measure belief across a range of themes and dimensions. This paper outlines two studies documenting the creation and psychometric properties of a novel measure assessing three different dimensions of belief across themes of politics, science, the paranormal, religion, and morality in UK samples ($n = 1,673$ total). Reliability estimates suggested good to excellent consistency (alpha > 0.8 per theme) with moderate to excellent reliability at 48 h (ICC = 0.61 –0.96) and 3.5 months (ICC = 0.61 –0.89). Factor analyses suggested good support for our five chosen themes of belief, suggesting they are distinct topic areas. Correlations across theme and dimension suggested dissociable characteristics within themes. These results have implications for 1. understanding the stability and relationship between themes of belief in a population and, 2. exploring how beliefs may change over time or as a result of an intervention. Full analysis code and data are available from the Open Science Framework (https://osf.io/hzvwr/).

# INTRODUCTION

Beliefs and values are central to both normal and abnormal psychology - for example, to understand political, religious, or moral behavior (*Graham, Haidt & Nosek, 2009*) or in the case of delusions (*Freeman, 2016*). Beliefs have been defined as 'a disposition to assent to, or otherwise act in accordance with some proposition' (*Sperber, 1996*) and have been

studied in a variety of ways. The attitudes and values of populations have long been studied using nationwide surveys (e.g., Ipsos and Gallup polls). While these give useful information about the distribution of beliefs and attitudes across a population, they have tended to treat beliefs as discrete items rather than as having different dimensions or components.

By contrast, the study of delusions as a type of abnormal belief in clinical psychology has explored dimensions and components of belief. For example, measures such as the Peter's Delusion Inventory (*Peters et al., 2004*) analyse delusional beliefs as including dimensions of distress, preoccupation, and conviction. Changes in these dimensions as a result of psychiatric intervention suggest that they are related but distinct characteristics (*So et al., 2014*).

The 'continuum' model of psychosis—which locates delusions and ordinary beliefs on a continuum—implies that dimensions of delusions should also be present in ordinary beliefs. Equally, the continuum model implies that dimensions of normal beliefs and their underlying social, cognitive, and neural processes can be linked to delusions. Indeed, paranoid and paranormal belief have been suggested as closely related to the content of delusion (*Brugger & Mohr, 2008*; *Bentall et al., 2001*). In addition, normal, socio-political beliefs such as conspiracy beliefs, while related to paranoia (*Imhoff & Lamberty, 2018*), are distinct entities in themselves. Research has shown them to be widespread (*Goertzel, 1994*; *Douglas et al., 2019*), driven by features such as partisanship (*Uscinski, Klofstad & Atkinson, 2016*), perceived threat, cultural essentialism (*Wood & Finlay, 2008*), and belief in unseen nefarious forces (*Oliver & Wood, 2014*). No study has systematically analysed the characteristics of a variety of *normal* beliefs as a basis for delusion. Understanding whether there are shared dimensions or characteristics across types of belief is important to help identify potential causal mechanisms of belief formation and maintenance.

Here we present a novel questionnaire aimed at 1. assessing multiple types or themes of propositional belief relating to the paranormal, religion, politics, morality, and science by bringing together previously dispersed themes (*Heath, Evans & Martin, 1994*; *Joseph & DiDuca, 2007*; *Tobacyk, 2004*; *Underwood & Teresi, 2002*) and, 2. assessing multiple dimensions of belief in relation to characteristics of agreement, self-relevance, and interest toward propositions. Dimensions were chosen based on observations that antipsychotics differentially reduce the preoccupation and distress of delusions (putatively one's interest in and perceived self-relevance of a belief) and conviction (agreement) in them (*So et al., 2014*). We aimed to test whether these dimensions were dissociable aspects of different types of propositional belief.

This article introduces and reports on the psychometric properties of the Beliefs and Values Inventory (BVI) using a large online general population sample specifically recruited on a platform with a large and diverse participant pool (*Peer et al., 2017*).

Two short studies drawing from five samples are reported, which are motivated by the following aims:

1. To understand the factor structure of our measure and identify whether beliefs were stable in the short (48 h) and long (3.5 months) term. This uses four separate online samples for cross sectional and longitudinal analysis.

2. To determine if these extracted items were stable in an independent sample. This was conducted using a confirmatory analysis to determine the weighting of the stable extracted factors. We also sought to understanding the association between agreement, self-relevance, and interest across and within themes. Samples from 1., and 2. were combined in a large composite sample.

## METHODS

Study 1 and 2 were both approved by King's College London research ethics committee (MRS-17/18-5956). All procedures were conducted in accordance with the committee's guidelines. Informed consent was obtained from all participants.

### Study 1
#### *Participants*
Prolific Academic (prolific.ac; hereafter Prolific), an online crowd-sourcing platform, was used to recruit all participants anonymously from the UK. The survey was hosted on Qualtrics (http://www.Qualtrics.com), an online survey and task platform. Each part of the study was put up online through Prolific until recruitment slowed to <2 responses/day or our participant quota was completed.

Our inclusion criteria were to recruit UK nationals over the age of 18 who were also fluent in English. Participants were asked not to participate if they ever had or currently had a mental health diagnosis.

We recruited at three time points:
1. The first sample recruited 1035 individuals anonymously through Prolific.
2. The second sample recalled 150 individuals from the first sample 48 h later using user IDs through Prolific.
3. The third sample recalled 69 individuals from the second sample (sample recruited at 48 h later) 3.5 months later using user IDs through Prolific.

Individuals who were part of the test-retest analyses in the first sample were excluded from the exploratory factor analysis.

#### *Beliefs and Values Inventory*
The BVI is a 55-item questionnaire (Appendix A) that contains propositional items across Political (10), Moral (10), Scientific (10), Paranormal (10), and Religious beliefs (10). There are an additional five items that act as attention check questions, for example, "Barack Obama was president of the USA."

Each theme contains 10 items, divided equally into two subthemes: more specific and more general propositions regarding the subject matter. For the purposes of reporting, we refer to these as 'specific' and 'general.' For example, a more specific science question is "Smoking contributes toward the development of lung cancer" and a more general science question is "Science will eventually be able to provide a more reliable account of human behaviour than literature, poetry, and art."

The division of themes into their subthemes was based upon previous phenomenological distinctions outlined in the literature. Specifically, items in the Moral theme were based

upon the moral (harm)/conventional (social) distinction outlined in Social Domain Theory (*Turiel, 1983*; *Turiel, 2015*)—these two classes of moral judgment are normally distinguished by age three (*Smetana, 1995*). The Moral Foundations Questionnaire was also used as a reference for the development of items in the morality theme (Haidt et al., 2009; Haidt et al., 2011). Political items were clustered by liberal/conservative dimensions and government responsibility and style (democratic/autocratic). These were constructed from the observations that political ideology tends to divide down liberal and conservative dimensions (*Smetana, 1995*; Haidt et al., 2009). Paranormal items were developed from concepts in existing questionnaires, for example, drawing from the Revised Paranormal Belief Scale (*Tobacyk, 1988*; *Joseph & DiDuca, 2007*). Beliefs toward religion and science were agreed upon through group consensus of the authors.

The general/specific distinction, while applicable to most themes, was chosen as a general indicator to delineate phenomenological clustering of items in a theme. This was for practical data handling and coding purposes more than a reflection of the content of a subtheme. For example, items in the Morality theme were divided by transgressional/conventional aspects of morality, but for the purposes of uniform data management were arbitrarily placed into general/specific subthemes, respectively.

Each item is further subdivided into three dimensions: agreement, self-relevance, and interest. Participants are asked to decide how much they agree with each statement, how relevant the statement is personally to them, and how interested they are in the statement. Participants score these dimensions on a visual scale from 0–10.

For the full listing of specific items see Appendix A.

## Analysis

All analysis was conducted in R (version 3.3.1, *R Core Team, 2013*) (full analysis code and data available from the Open Science Framework: https://osf.io/hzvwr/). R is an open source statistical software with robust packages to cover a range of analyses. All visualisation was conducted using the *ggplot2* package (*Wickham, 2016*).

An exploratory analysis on the demographics, reliability (Cronbach's alpha), exploratory factor structure, and intra-class coefficients were performed.

## Data cleaning and scoring

Participants were removed if they did not at least answer '6' (agree) on the control questions.

We coded items based upon their theme (W), theme subsection (X), question number (Y), and dimension (Z) as follows: "W_XY_Z", where Y is a number from 1–5. For example, the 3rd politics question (agreement dimensions) from its 'specific' subsection would be "Pol_S3_A".

Specifically, codes for items were:

*Theme*

– Pol = Politics
– M = Morality
– R = Religion
– P = Paranormal

– S = Science

*Theme subsection*

– G = General
– S = Specific

Agreement score for questions "Pol_G3_A", "Pol_G5_A", "Pol_S5_A", "M_G5_A", "M_S2_A", "M_S4_A", "R_G4_A", "S_G1_A", and "S_S4_A" were reverse coded so that all agreement in each theme moved toward the same overall concept.

Therefore, higher ratings in each theme for the agreement dimension indicated:

- General Politics –Government responsibility and style (the use of government and taxes to benefit the population and address wellbeing).
- Specific Politics –High agreement with liberal values addressed using specific policy.
- General Morality –High agreement with acceptability of moral transgressions (harm to others in absence of discovery).
- Specific Morality –High agreement with acceptability of conventional transgressions (in absence of discovery).
- General Science –High agreement with the power of science as a tool for establishing reliable knowledge.
- Specific Science –High agreement with current evidence for specific scientific questions.
- General Religion –High agreement with propositions expressing religious views about the nature of reality.
- Specific Religion –High agreement with religion as a tool for good in the world and society.
- Paranormal (both general and specific) –High agreement with superstitious concepts (e.g., "The number 13 is unlucky") and magical thinking (e.g., "Crystals can be used for healing").

*Dimension*

- Agreement (A)
- Interest (I)
- Self-Relevance (R)

High scores on the interest and self-relevance dimensions indicate high interest and high self-relevance, respectively, across all themes.

### Demographics

Frequencies were calculated for the age, sex, religion, mother's religion, father's religion, ethnicity, education, and political orientation. Religion, ethnicity, religion, and education demographic options were based upon the UK census (*Office for National Statistics, National Records of Scotland & Northern Ireland Statistics and Research Agency, 2016*). Parent's religion was included as they may relate to the development of beliefs in the participant.

### Descriptive statistics

Total answer density scores were calculated for each dimension for the original and all retest samples. This gives a visual distribution of the sum of item scores for each theme across each sample.

### Reliability (Cronbach's Alpha)

Cronbach's alpha was used to compute the internal reliability for the entire questionnaire, and each theme calculated separately for each dimension. Calculations were completed with the *Psych* package (v.1.8.4; *Revelle, 2018*) for R.

In the case of our questionnaire, alpha values also give a reliable representation of the volatility of beliefs across the group—in essence, how frequently participants answer in the same way for agreement, interest, and reliability across themes in the group.

### Test-retest intraclass correlation coefficients

Intraclass Correlation Coefficients (ICC) compute the relative similarity of quantities within a group, members of which have been given identical measurements (*Shrout & Fleiss, 1979*; *Koch, 2006*). This provides a metric as to how consistent a group of participants' answers are between two time points, in this case, over 48 h and over 3.5 months. ICC analysis has been suggested as a robust measure of test-retest reliability when comparing factor structures and psychometric tools (*Weir, 2005*). An ICC of between 0.5 and 0.75 is considered "moderate", between 0.75 and 0.9 is considered "good", and an ICC over 0.9 is considered excellent (*Koo & Li, 2016*).

Item dimensions grouped by each theme were summed and input into an ICC analysis using the *Psych* package. ICC analyses were conducted to assess the stability of answers over 48-h and 3.5-month periods. The 48-h analysis used ID matched participants from baseline and 48-h time points. The 3.5-month analysis used ID matched participants from baseline and 3.5-month time points—these were recruited from participant IDs who had taken part in the 48-h analysis to enable ICC comparison between samples.

Item totals for each dimension and theme were summed for each participant and compared between time points. For example, agreement with scientific statements were summed for participants at 0 h and 3.5 h and analysed for consistency. We used Model 3 (3, 1), which considers our population as the only population of interest and further counts one single measure per time point (*Koo & Li, 2016*; *Shrout & Fleiss, 1979*).

### Exploratory factor analysis (EFA)

We selected five themes to comprise the BVI, each containing a number of theme-relevant items. An EFA was used to support our presupposed structure in the BVI. This also served to identify items which may not contribute to the theme reliably.

In an EFA, factors are extracted which represent putative latent variables underlying a group of items. These latent variables are then able to be tested in a confirmatory analysis with a new sample.

Factor analysis was conducted as opposed to a principal component analysis because we aimed to determine to what degree each item within a theme contributed to the latent variable of belief. We used the *Psych* package (v.1.8.4; *Revelle, 2018*) for R.

Parallel analysis was first conducted to calculate the appropriate number of eigenvectors to extract. Appropriate number of factors were determined upon viewing the scree plot (*Cattell, 1966*).

EFA was applied to the dimensions Agreement, Self-Relevance, and Interest to determine overall loading across themes. An exploratory orthogonal (varimax) factor analysis was conducted, which allows for the factors to not correlate. A threshold of 0.4 was applied to factor loadings.

## Study 2
### Participants
Prolific.ac was used to recruit all participants anonymously from the general population in the UK.

Inclusion Criteria:

- UK National
- Over the age of 18
- Do not have or never have had a diagnosis of a mental health condition
- Prolific.ac approval rate over 80%

We recruited 488 anonymous participants at one time-point.

The original BVI-55 was used (before item removal), however for confirmatory factor analysis, only the items shown to be significantly weighted to a factor were retained for analysis.

The extra items were used to collate large sample size associations.

### Analysis
All analysis was conducted in R (version 3.3.1; *R Core Team, 2013*) (full analysis code and data available from the Open Science Framework: https://osf.io/hzvwr/). Both are open source statistical software with robust packages to cover a range of analyses. All visualisation was conducted using ggplot2 (*Wickham, 2016*).

A confirmatory factor analysis of the factor structure obtained in **Study 1** was conducted.

### Data cleaning and scoring
All data cleaning procedures for **Study 2** are identical to those performed in **Study 1**. For further details please see our script.

### Descriptive statistics
Total answer density scores were calculated for each dimension for the confirmatory sample. This gives a visual distribution of the sum of item scores for each theme, dimension, and sample.

### Exploratory associations
Samples from **Study 1** (original - $n = 736$; retest1 - $n = 98$) and **Study 2** (confirmatory $-n = 381$) were combined to form a new large sample ($n = 1215$). No significant mean differences were found between samples across themes and item dimensions.

We conducted Pearson *r* correlations to understand the associations of all 55 item totals across themes within dimensions.

### Cronbach's Alpha

Cronbach's alpha was used to compute the cross-sectional reliability in the confirmatory ($n = 381$) sample using items retained from the EFA in **Study 1**. This gives a metric to identify the coherence of answers across the entire questionnaire and its component themes. We used the *Psych* package (v.1.8.4; *Revelle, 2018*).

### ICC analysis (means weighted by factor)

Item dimensions grouped by each factor (determined from the pattern matrix output in **Study 1**) were multiplied by their factor loading to generate a weighted score, summed and input into an Interclass Correlation Coefficient (ICC) analysis using the *Psych* package in R to assess the stability of answers over 48 h and 3.5 months.

### Confirmatory Factor Analysis (CFA)

A CFA was run to determine the strength of loading of each item onto factors identified in the EFA. This also provides a metric to determine the interaction of the factors identified in **Study 1**.

Using the results from **Study 1**'s exploratory factor analysis, six CFAs were run to determine the degree to which the extracted items under each theme across agreement, interest, and self-relevance fitted together in:

1. 5-factor model, where politics and morality are treated as separate latent variables.
2. 4-factor model, where politics and morality are treated as a single latent variable.

The model was specified using items extracted under each theme using the outputs from the EFA across agreement, interest, and self-relevance. Items were grouped into a theme if they loaded independently onto a factor without cross-loading.

All items were coded as numerical continuous variables (from 0 –10).

A confirmatory factor analysis was conducted using the R package 'lavaan' ('cfa' and 'sem' function; *Rosseel, 2012*).

The package 'semPlot' was used to visualise the results using the 'semPaths' function.

Maximum likelihood estimation was used. The latent factors were standardised allowing free estimation of all factor loadings.

Agreement, Self-Relevance, and Interest were fitted separately across models (1) and (2).

Networks of observed and implied items within each factor model were created using the "semCors" function in semPlot. These used LASSO estimation algorithms.

## RESULTS

### Study 1

After cleaning (see https://osf.io/hzvwr/), four samples were obtained (see Fig. 1):

- $N = 736$ (Time point 0 sample excluding test-retest participants)
- $N = 98$ (Time point 0 sample test-retest participants)

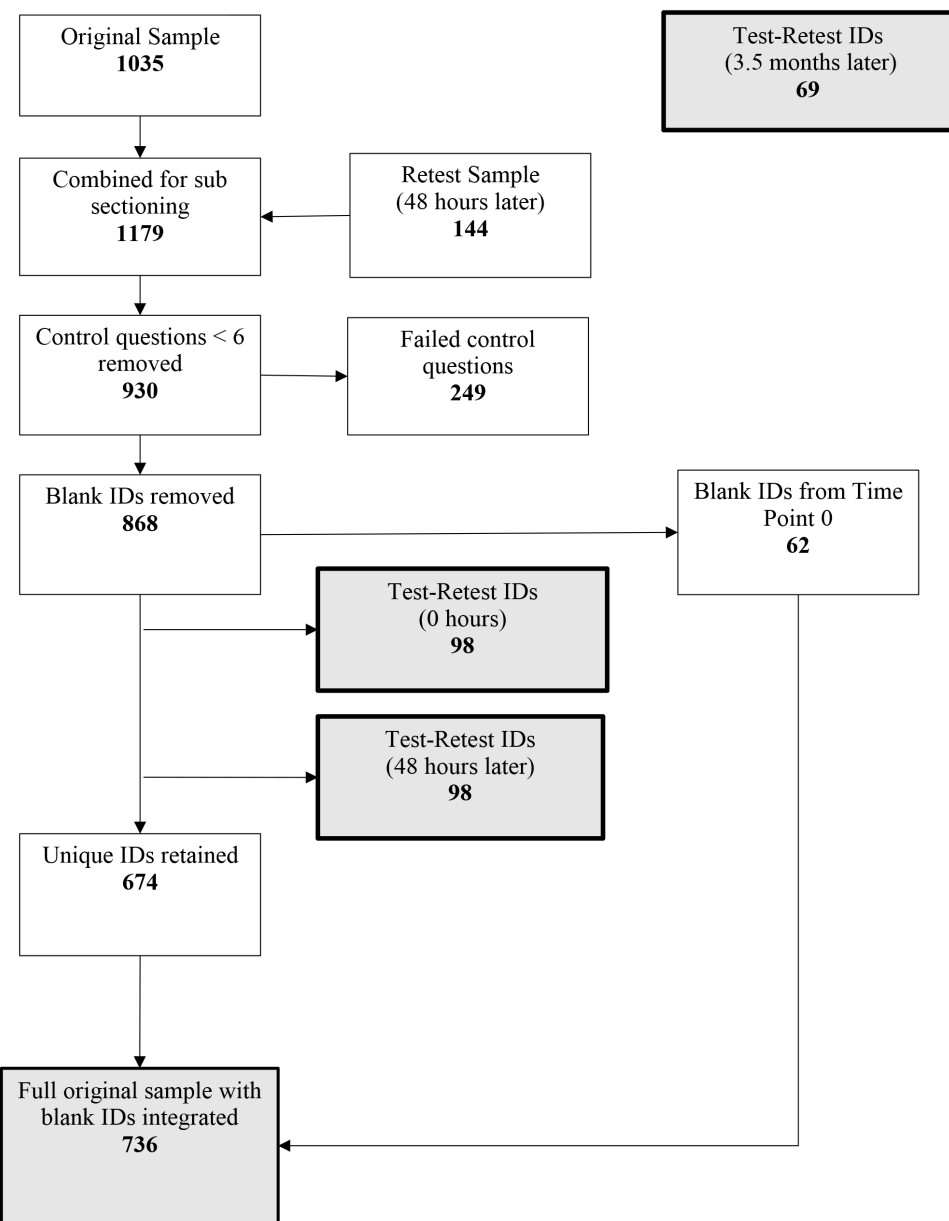

**Figure 1** **Path diagram to illustrate data cleaning, control, and sub-setting of the original sample ($n = 736$), retest 1 (0 h, $n = 98$), retest 2 (48 h, $n = 98$) and retest 3 sample ($n = 69$).** Grey boxes indicate final samples used for analysis. Blank IDs from the original sample were removed. This ensured that only blank IDs were from '0 h' participants and not from the retest sample (48 h). Therefore, they were able to be reintegrated after the separation of the retest sample.

- $N = 98$ (Time point 48 h sample test-retest participants)
- $N = 69$ (Time point 3.5 months sample test-retest participants)

The Time Point 0 and Time Point 48 h samples were combined and ordered by submission to assess the frequency of independent identification numbers (ID). Frequencies

**Table 1  Demographics of the original sample ($n = 736$) excluding those used for ICC analysis.**

| Category | | | | | | | Frequency ($n = 736$) | | |
|---|---|---|---|---|---|---|---|---|---|
| Sex | | | | Male | | | | Female | |
| | | | | 286 | | | | 449 | |

| Age | 18–25 | 26–30 | 31–35 | 36–40 | 41–45 | 46–50 | 51–55 | 56–60 | 61–65 | 65+ |
|---|---|---|---|---|---|---|---|---|---|---|
| | 110 | 135 | 104 | 101 | 68 | 80 | 61 | 32 | 28 | 16 |

| Political Affiliation | Labour | Conservative | Green | SNP | DUP | UKIP | Liberal Democrat | Other | None |
|---|---|---|---|---|---|---|---|---|---|
| | 287 | 128 | 30 | 18 | 5 | 19 | 40 | 3 | 206 |

| Religion | Christian | Islamic | Jewish | Hindu | Buddhist | Sikh | Spiritualist | Atheist | Agnostic | Other |
|---|---|---|---|---|---|---|---|---|---|---|
| Participant | 313 | 17 | 5 | 2 | 4 | 5 | 9 | 256 | 85 | 40 |
| Mother | 505 | 16 | 7 | 3 | 5 | 5 | 6 | 118 | 46 | 25 |
| Father | 409 | 18 | 7 | 5 | 5 | 5 | 3 | 182 | 61 | 45 |

| Education | Primary | Secondary | A levels/ College | Undergraduate | Postgraduate | Masters | Doctorate |
|---|---|---|---|---|---|---|---|
| | 2 | 103 | 180 | 317 | 38 | 80 | 16 |

of independent IDs over 2 (retest participants) were then removed from the sample to isolate participants that had completed the questionnaire at the two time-points. Participant IDs were then ordered by submission and separated to generate samples at 0 h and 48 h. The demographics of the sample at time point 0 can be found in Table 1.

Blank IDs (IDs where participants had not recorded their prolific participant numbers) were removed from the original sample to prevent unknown duplicates being detected between "Test-Retest IDs" at time points 0 and 48 h when sub setting in R.

Retest participants collected 3.5 months later were ID matched to their time 0 answers. 3.5-month participants were treated as a separate sample and compared to their time 0 answers in test-retest analysis. See Fig. 1 for a visual description.

### Demographics
### Descriptive statistics
The distribution of answers (density) within item dimensions (agreement, relevance and interest) across themes (paranormal, moral, politics, science, and religion) of the BVI-55 were calculated for the original, and all three retest samples.

No significant differences in mean were found between the original and retest 1 sample (0 h or baseline) samples.

Graphical histogram comparisons of original and retest samples can be found in **Study 2.**

### Reliability –Cronbach's Alpha
The BVI as a whole showed good internal reliability (alpha of 0.95; 95% CI [0.95–0.96]). By theme, Cronbach's alpha scores were reported as Science (0.87), Paranormal (0.96),

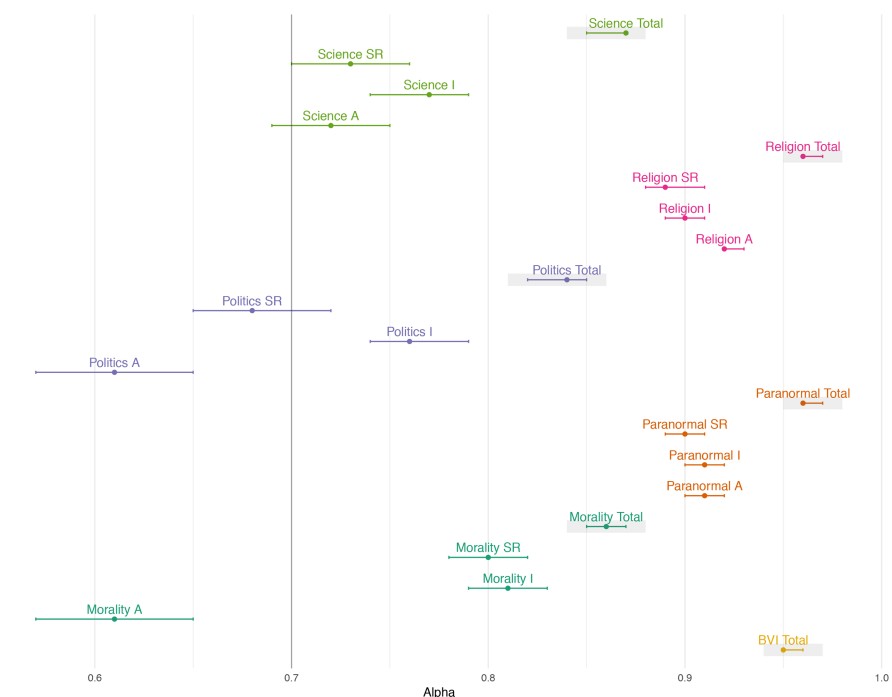

**Figure 2** **Cronbach's alpha for all BVI individual item dimensions by theme.** Bars represent 95% confidence intervals. BVI, Beliefs and Values Inventory; Total, Total items across all dimensions; A, Agreement dimension; R, Self-Relevance dimension; I, Interest dimension. Grey boxes highlight total scores. We found that the total measures of themes were very stable, with their respective dimensions varying to different degrees.

Politics (0.84), Religion (0.96), and Morality (0.86). All item dimensions and theme had an alpha of above 0.7 except Morality-Agreement (0.61), Politics-Agreement (0.61), and Politics-Self Relevance (0.65).

To understand whether scores below 0.7 were due to issues with summation across general and specific subthemes, Morality-Agreement, Politics-Agreement, and Politics-Self Relevance were split by their subthemes.

Splitting themes by their subthemes did not improve alpha:

- Morality-Agreement (Moral transgressions) = 0.46 (95% CI [0.40–0.52])
- Morality-Agreement (Conventional transgressions) = 0.54 (95% CI [0.49–0.59])
- Politics-Agreement (Socialist principles) = 0.45 (95% CI [0.39–0.51])
- Politics-Agreement (Liberal values) = 0.53 (95% CI [0.48–0.58])
- Politics- Self Relevance (Socialist principles) = 0.56 (95% CI [0.41–0.61])
- Politics- Self Relevance (Liberal values) = 0.61 (95% CI [0.57–0.66])

Figure 2 visually represents the raw alpha and 95% confidence interval between BVI total, theme totals, and themes by their item dimensions.

### ICC analysis

We aimed to establish whether agreement, interest, and self-relevance toward themes of belief in the BVI were stable over a 48-h and 3.5-month period.

After cleaning we had two samples:

1. Ninety-eight ID matched participants to check stability over a 48-h period. This used participants from Time 0 h Retest and Time 48 h Retest participants.
2. Sixty-nine ID matched participants were used to check stability over a 3.5-month period. This used participants from Time 0 h Retest and Time 3.5 Month Retest participants. Fewer participants were in this group because of attrition between recruitment via Prolific.ac between 48 h and 3.5 months.

ICCs across all dimensions, themes, and time samples were generally good to excellent (>0.75).

Exceptions were:

1. At 48 h
   The self-relevance dimension and agreement dimension of the Morality theme scored moderate reliability, scoring 0.68 and 0.61 respectively.
2. At 3.5 months
   All dimensions of morality were of moderate reliability, all scoring between 0.61 and 0.69. Self-relevance dimensions for Politics and Science themes scored moderate reliability, scoring 0.69 and 0.71, respectively.

Figure 3 demonstrates the ICC (3,1) for agreement, interest, and self-relevance item dimensions across themes over 48 h and 3.5 months with their respective 95% CI. Weighted means are discussed later.

See Appendix A for all items in each theme of the BVI used in Study 1. See Appendix D for full statistical output and ICC breakdown.

### Exploratory factor analysis

Exploratory Factor Analysis (Orthogonal, Varimax rotation) was run with a parallel analysis. This was to determine the number of factors and items to extract for a confirmatory analysis of the BVI. This was conducted for the total questionnaire (no dimensional partitioning) and all three BVI dimensions individually.

### Total BVI

A solution of 10 factors was considered adequate upon observation of the scree plot and cross-checking with the *Psych* package "nfactor" function to compute optimal eigenvalue decomposition (*Revelle, 2018*).

Factors 1–5 explained 0.77 of the proportion across factors (see Table 2).

Summary of items that loaded onto each factor across the BVI can be found in Appendix B.

### Agreement

A solution of seven factors was considered adequate upon observation of the scree plot and cross-checking with the *Psych* package "nfactor" function to compute optimal eigenvalue decomposition (*Revelle, 2018*).

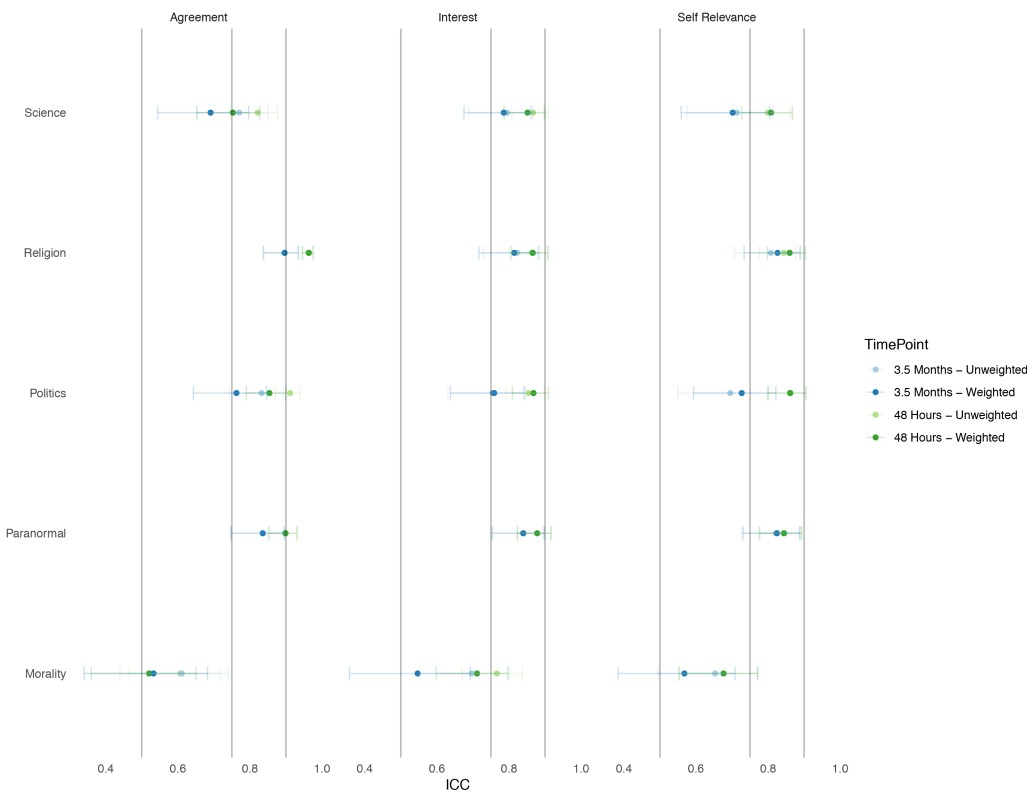

**Figure 3** ICC (3,1) scores (95% CI) over a 48-hour period ($n = 98$), and 3.5- month period ($n = 69$) across themes within item dimensions for weighted and unweighted means. Bars represent standard error. Vertical lines are inserted to denote ICC cut off at 0.50 (moderate), 0.75 (good) and 0.90 (excellent). Overall we found moderate to excellent evidence that beliefs are stable over 48 hours and 3.5 months.

**Table 2** Variance and Sum of Squares loading for each extracted factor across all themes of the BVI. Figures recorded to two decimal points.

| Statistic | Factor | | | | | | | | | |
|---|---|---|---|---|---|---|---|---|---|---|
| | 1 | 2 | 3 | 4 | 5 | 6 | 7 | 8 | 9 | 10 |
| SS loading | 24.10 | 22.33 | 11.61 | 11.51 | 10.00 | 5.89 | 5.18 | 4.72 | 4.27 | 3.15 |
| Proportion variance | 0.12 | 0.11 | 0.06 | 0.06 | 0.05 | 0.03 | 0.03 | 0.02 | 0.02 | 0.02 |
| Cumulative variance | 0.12 | 0.24 | 0.30 | 0.36 | 0.41 | 0.44 | 0.46 | 0.49 | 0.51 | 0.53 |
| Proportion explained | 0.23 | 0.22 | 0.11 | 0.11 | 0.10 | 0.06 | 0.05 | 0.05 | 0.04 | 0.03 |
| Cumulative proportion | 0.23 | 0.45 | 0.56 | 0.68 | 0.77 | 0.83 | 0.88 | 0.93 | 0.97 | 1.0 |

Factors 1 and 2 explained 32% and 31% of the proportion across factors respectively, with the rest of the factors contributing less than 10%.

Full loadings and outputs can be found in Appendix B.

### Self-relevance

A solution of seven factors was considered adequate upon observation of the scree plot and cross-checking with the *Psych* package "nfactor" function to compute optimal eigenvalue decomposition (*Revelle, 2018*).

Factors 1–4 explained 27%, 23%, 14%, and 13% of the proportion respectively, with the rest of the factors contributing less than 10%.

Full loadings and outputs can be found in Appendix B.

### Interest

A solution of seven factors was considered adequate upon observation of the scree plot and cross-checking with the *Psych* package "nfactor" function to compute optimal eigenvalue decomposition (*Revelle, 2018*).

Factors 1–5 explained 27%, 21%, 15%, 14%, and 12% of the proportion across factors, respectively, with the rest of the factors contributing less than 10%.

Full loadings and outputs can be found in Appendix B.

### Item selection

Following the exploratory factor analysis, all items that did not load onto any factor or cross-loaded with another item were removed for confirmatory analysis.

If a proposition from one dimension loaded onto a factor, but a different dimension from the same proposition didn't load onto any factor in the analysis, that proposition was still retained in the revised questionnaire. For example, item 'S_S1_I' loaded onto a factor for the *interest* item dimension, but was absent from the *agreement* analysis, so was retained.

Control questions were all retained.

Forty-two items from the BVI were retained for confirmatory analysis (10 paranormal, 10 religious, six science, five politics, six morality, and five control questions).

All retained items can be found in Appendix C (BVI-42).

## Study 2
### Participants

Following data cleaning by control questions (using the same procedure for **Study 1**) we were left with 381 participants. All participants in the cleaned sample were then used for all further analyses

### Descriptive statistics

The distribution of answer total (density) within item dimensions (agreement, relevance and interest) across themes (paranormal, moral, politics, science, and religion) of the BVI-55 were calculated.

Figure 4 demonstrates the density across agreement, self-relevance and interest across all five samples, including the confirmatory sample.

### Cronbach's Alpha

Cronbach's alpha was used to test the cross-sectional reliability within the confirmatory sample ($n = 381$) across all the entire questionnaire and themes (Morality, Paranormal, Politics, Religion, Science) with items retained from the EFA in **Study 1**.

The BVI as a whole reached a very high reliability (0.96; 95% CI [0.95–0.96])

When divided by theme alone, alpha scores maintained very high scores. This was true for Science (0.87), Paranormal (0.96), Politics (0.75), Religion (0.96), and Morality (0.86).

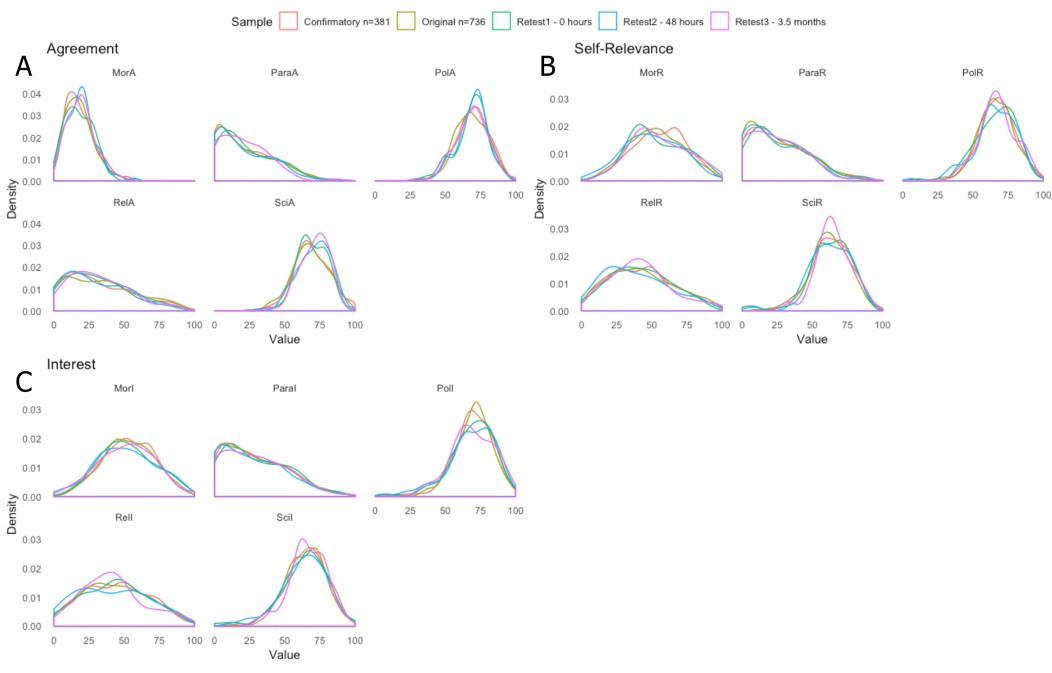

**Figure 4** Density of answer totals within themes and faceted by Agreement (A), Self-Relevance (B) and Interest (C) dimensions, across original ($n = 736$), retest 1 (0 h; $n = 98$), retest 2 (48 h; $n = 98$), retest 3 (3.5 months; $n = 69$) and confirmatory samples ($n = 381$). Mor, Morality; Para, Paranormal; Pol, Politics; Rel, Religion; Sci, Science. A, Agreement; R, Relevance; I, Interest. No significant differences in mean scores were found between the original, retest1 and confirmatory samples.

### Exploratory associations

Within dimensions, self-relevance and interest were positively and significantly ($p < 0.001$) associated for all themes. Morality and politics, religion and paranormal, and politics and science had particularly strong associations ($r > 0.5$).

Within the agreement dimension, morality was not strongly associated with any other theme. Religion and paranormal, and science and politics themes were moderate to strong and positively correlated ($r > 0.4$). Religion and science, and science and paranormal items showed a moderate to strong negative association ($r < 0.4$).

Figure 5 gives a visual depiction of all associations within dimensions and themes.

We aimed to establish whether agreement, interest, and self-relevance toward themes of belief in the BVI were stable over a 48-h and 3.5-month period when items were reduced and summed by their weighted mean within factors suggested by the EFA in **Study 1**.

After cleaning we had two samples:

1. Ninety-eight ID matched participants to check stability over a 48-h period (0-h retest and 48-h retest).
2. Sixty-nine ID matched participants were used to check stability over a 3.5-month period (0-h retest and 3.5-month retest).

ICCs across all dimensions, themes, and time, samples were generally good to excellent (>0.75).

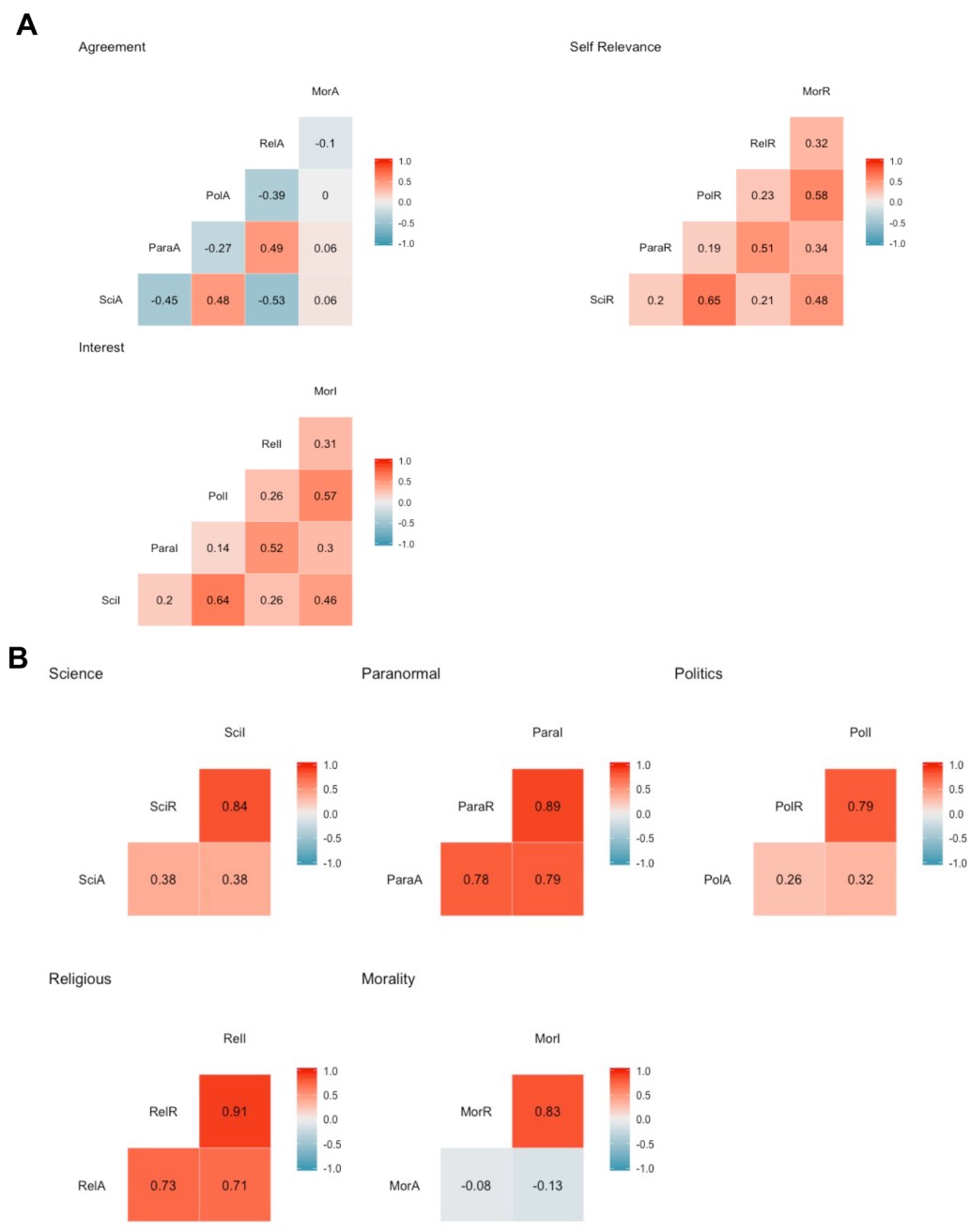

**Figure 5 Pearson r correlation matrices.** (A) Associations within dimensions across themes. (B) Associations within themes across dimensions. Both include all items before exclusion. All correlations are rounded to two decimal points. This used samples from Mor, Morality; Para, Paranormal; Pol, Politics; Rel, Religion; Sci, Science. A, Agreement; I, Interest; R, Self- Relevance. Self-relevance and interest dimensions were positively and significantly correlated across all themes. For the agreement dimensions, there was high variation between themes. The correlations between agreement and interest, and agreement and self-relevance were significantly lower than the correlation between interest and self-relevance for all themes ($ps < 0.001$).

Exceptions were:
1. At 48 h
   All dimensions within the Morality factor were between 0.52 and 0.71.
2. At 3.5 months
   All dimensions within the Morality factor, the self-relevance dimension of the Politics factor, and agreement and self-relevance dimensions of the Science factor were between 0.53 and 0.72.

Figure 3 demonstrates the ICC (3,1) for agreement, interest, and self-relevance item dimensions across factors (weighted) and themes (unweighted –from **Study 1**) over 48 h and 3.5 months with their respective 95% CI.

See Appendix E for full statistical output and ICC breakdown.

### Confirmatory Factor Analysis (CFA)
### Agreement
A 5-factor fit was acceptable (Tucker-Lewis Index = .84; RMSEA = .075–90%CI [.070–.080]) and fitted the data significantly better than a 4-factor model ($\chi^2(4) = 117.2$, $p < 0.001$).

Significant positive factor loadings were found with all items but one (0.57 –2.73) with only one politics item reaching a low of 0.14.

### Interest
A 5-factor fit was acceptable (Tucker-Lewis Index = .86; RMSEA = .059–90%CI [.055–.063]) and fitted the data significantly better than a 4-factor model ($\chi^2(4) = 140.54$, $p < 0.001$).

Significant positive factor loadings were found with all items (1.13 –2.87).

### Self-relevance
A 5-factor fit was acceptable (Tucker-Lewis Index = .86; RMSEA = .057–90%CI [.053–.062]) and fitted the data significantly better than a 4-factor model ($\chi^2(4) = 126.2$, $p < 0.001$).

Significant positive factor loadings were found with all items ($\beta = 0.92 – 2.87$).

Figure 6 demonstrates the confirmatory factor models for each dimension, including inter-factor associations.

## DISCUSSION

The Beliefs and Values Inventory (BVI) was developed to measure dimensions of different types of belief found in the general population. Specifically, it aims to measure dimensions of agreement, self-relevance, and interest towards propositions from paranormal, religion, politics, morality, and science themes. It was constructed with the dimensions drawn from literature on normal and pathological belief formation.

Cronbach's alpha suggested that theme summary scores and the total BVI scores were stable (when agreement, self-relevance and interest were combined), but dimensions analysed individually varied somewhat. All dimensions of the paranormal and religion theme were highly reliable, and all dimensions of the science theme were satisfactorily

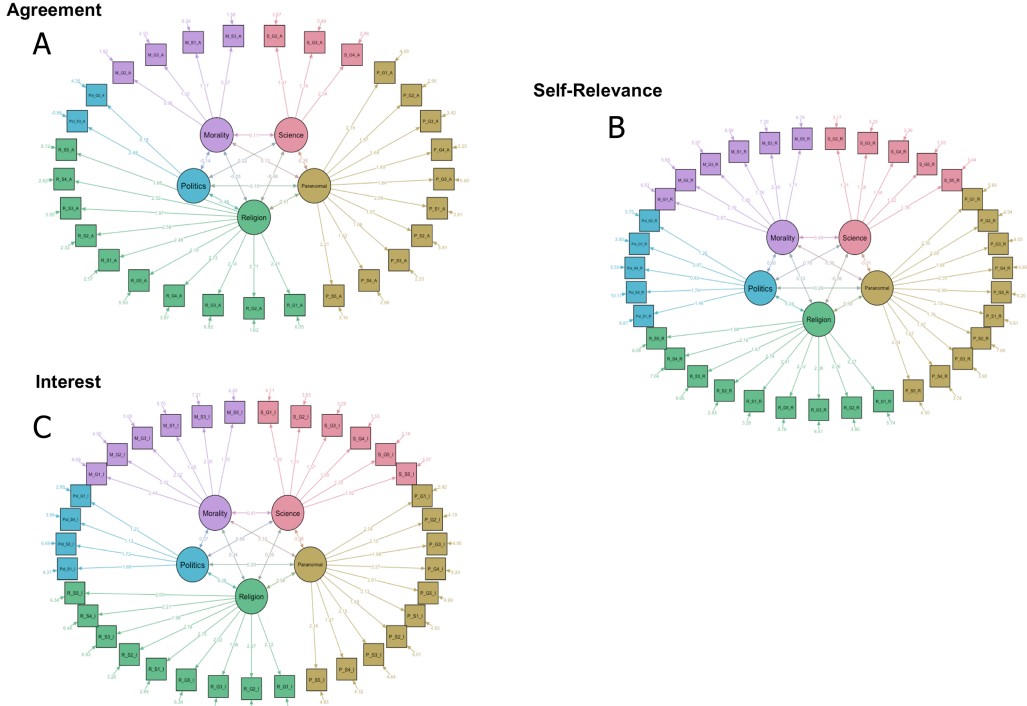

**Figure 6  Agreement (A), Self Relevance (B), and Interest (C) confirmatory factor models with a 5-factor solution.** Each factor has the same names as the themes originally devised in the BVI-55 as each factor included items that were originally in the pre-named themes devised. Numbers represent regression (b) coefficients. All items included for each dimension model scored > 0.4 loading in the exploratory factor analysis in Study 1.

reliable. Agreement for both politics and morality scored lower than other themes. This was perhaps in part due to the variance in item domains that comprised each theme. Participants may agree strongly with one item (e.g., belief in abortion as a positive influence) but not others (e.g., that public services are necessary for the greater good), and this disparity was not uniform across the group.

Test-retest ICC analysis confirmed that beliefs are stable over a 48-h and 3.5-month period, although belief in moral concepts are less stable. As with Cronbach's alpha scores, paranormal and religion themes were most stable. The science and politics themes were also stable. This indicates that the contrast between reliability scores (Cronbach's alpha and ICC) is reflective of items being answered in a very similar manner over time despite variance within the theme itself. Morality was the least stable, but still achieved moderate ICC scores. This is perhaps unsurprising given the similar test-retest correlation of questions relating to moral harm in a previous sample (*Graham et al., 2011*). We suggest moderate ICC scores are found because of the changeable nature of moral beliefs over time; for example, political or life events may cause attitudes toward a proposition to alter. In addition, despite stable politics and science ICC scores, wide confidence intervals may suggests that dimensions might be less stable because of current political or life developments (even within 48 h).

Exploratory and confirmatory analyses suggest that a 5-factor model of the BVI-42 is supported, with factors consistent with pre-defined themes after item reduction. Notably, when performing an exploratory factor analysis on the entire BVI-55, items fitted into five factors, mainly segregated into their five themes (see Appendix B). This included a mix of agreement, self-relevance and interest dimensions for each factor, meaning that the confirmatory factor analysis on each dimension contained a different number of items.

Given the strong coefficient scores from the confirmatory factor analysis the BVI-55 or BVI-42 could be used, however this is not straight forward. Taking into consideration that the reliability estimates were considerably higher in the BVI-55, we suggest using the BVI in its full 55 item form using the summary of each theme and dimension for total scores. Theoretically, this is also supported by the fact that items that do not load strongly onto one factor may not necessarily be irrelevant. Instead they are reflective of the broad range of phenomenological content within each theme. Therefore, we would expect all items to be useful when examining change in belief over time in a single individual. The intended use of this questionnaire is as a single, composite measure to observe alterations in belief over time. Consequently, summary scores for each theme will encompass all the necessary variation of each theme adequately. This means that it is unnecessary to remove items that are orthogonal to the latent factors underlying particular sets of items within the theme.

Correlational analyses suggest that beliefs are dissociable by dimension and theme to varying degrees. When looking within themes, dimensions are dissociable –both interest and self-relevance are highly associated, and their relationship with agreement is equivalent, but significantly weaker. In addition to observing within-dimension correlations, it is clear that self-relevance and interest are separate components to agreement which is highly variable in its association between themes. It is notable to point out that agreement between themes are associated as one may expect, e.g., paranormal and religious items are positively associated, as are liberal values (politics) with belief in science as a tool for enquiry. However, it is striking that morality did not correlate with agreement from the other themes. We suggest that within the range of scores in our large population this is perhaps due to its independence from other beliefs—religious or liberal status does not necessarily have a uniform impact on moral values. In combination within the exploratory factor analysis, we suggest that themes are distinct, but associated entities, as are dimensions within a particular theme.

Despite the similarity between associations of self-relevance and interest within themes, we suggest keeping them as separate dimensions. This is based on theoretical grounds. In a healthy population these constructs appear to be highly coherent, perhaps the same. However, we do not yet know how this measure fares in clinical populations or intervention designs. The Peter's Delusion Inventory (PDI; *Peters et al., 2004* has suggested similar trends. Both the original study (*Peters et al., 2004*) and more recent studies (*Sisti et al., 2012*) using the questionnaire have found its dimensions of distress and preoccupation to be highly related in healthy and clinical populations, and dissociable from the conviction dimension. However, in some populations these dimensions are dissociable (*Peters et al., 1999*). Likewise, self-relevance and interest dimensions are theoretically separable but are

highly associated. It remains possible that this is not the case in different populations or under different conditions.

There are a few limitations to note. We have not yet tested this measure in a clinical population. The dimensional structure of our measure may be variable when used in those with a need for care, or indeed when participants are part of a drug intervention. Additionally, our questionnaire has not yet been compared with other measures that purport to measure the same thematic constructs. Such comparisons are ongoing. Finally, our morality theme appears to be the least stable relative to the other themes in our measure as demonstrated by lower ICC, Cronbach's alpha, and correlational analyses. We therefore suggest that the most statistically stable form of the questionnaire may be to use summary scores of paranormal, political, science, religion, and control items, with morality items being used if it is of theoretical interest. We hope that future studies using the questionnaire in clinical populations, as part of an intervention, and alongside other measures will test the variability and validity of the BVI, in addition to the validity and reliability of the morality dimension.

The reliability of the BVI show that it has utility in the assessment of belief over time. Additionally, it can be used in cohort research alongside measures such as the PDI to test if certain beliefs are more closely aligned with delusions (*Bentall et al., 2001*). Converging evidence on the stability of belief over time and within different individuals will help understand mechanisms of belief change, and whether certain cognitive components are more involved in the development of inflexible beliefs.

## CONCLUSION

The Beliefs and Values Inventory (BVI) is a multidimensional tool assessing belief agreement, interest, and self-relevance. Correlational analyses confirm that belief is comprised of multiple dimensions. Results additionally suggest that beliefs can be stable over a 48 h and 3.5-month period across dimensions. In regard to politics and morality themes, answers should be interpreted with more caution when considering the agreement dimension and items may need to be analysed individually for more specific conclusions. Through intervention, the BVI may reveal differential changes in dimensions of beliefs in a population—whether clinical or otherwise.

## ACKNOWLEDGEMENTS

We would like to thank Sara DeSimoni for her work constructing the prototype BVI. We would also like to thank Georgianna Adams for her useful discussion around beliefs about physical therapy during the construction of this questionnaire.

### Funding

Joseph M. Barnby is supported by the UK Medical Research Council (MR/N013700/1) and King's College London member of the MRC Doctoral Training Partnership in

Biomedical Sciences. Vaughan Bell is supported by a Wellcome Trust Seed Award in Science (200589/Z/16/Z). The funders had no role in study design, data collection and analysis, decision to publish, or preparation of the manuscript.

### Grant Disclosures

The following grant information was disclosed by the authors:
UK Medical Research Council: MR/N013700/1.
King's College London member of the MRC Doctoral Training Partnership in Biomedical Sciences.
Wellcome Trust Seed Award in Science: 200589/Z/16/Z.

### Competing Interests

The authors declare there are no competing interests.

### Author Contributions

- Joseph M. Barnby conceived and designed the experiments, performed the experiments, analyzed the data, prepared figures and/or tables, authored or reviewed drafts of the paper, approved the final draft.
- Vaughan Bell, Mitul A. Mehta and Quinton Deeley conceived and designed the experiments, authored or reviewed drafts of the paper, approved the final draft.
- Luke Sheridan Rains approved the final draft, developed the prototype Beliefs and Values Inventory.

### Human Ethics

The following information was supplied relating to ethical approvals (i.e., approving body and any reference numbers):

Kings College London Research Ethics Commitee granted Ethical approval to carry out the study (Ref: MRS-17/18-5956).

### Data Availability

All data, preprints, scripts, and the BVI-55 are available at the Open Science Framework:

Barnby, Joseph M, Vaughan Bell, Luke Sheridan-Rains, Mitul Mehta, and Quinton Deeley. 2019. "Beliefs and Values Inventory." OSF. January 22. doi: 10.17605/OSF.IO/HZVWR.

### Supplemental Information

Supplemental information for this article can be found online at http://dx.doi.org/10.7717/peerj.6819#supplemental-information.

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
