# Peer review of "Beliefs are multidimensional and vary in stability over time - psychometric properties of the Beliefs and Values Inventory (BVI)"

_PeerJ, doi:10.7717/peerj.6819_

## Round 0.1 · original submission · Major Revisions

· Academic Editor

Major Revisions

Your manuscript entitled "Beliefs are multidimensional and vary in stability over time - psychometric properties of the Beliefs and Values Inventory (BVI)" has now been seen by 2 referees. You will see from their comments below that while they find your work of interest, some important points are raised. We are interested in the possibility of publishing your study, but would like to consider your response to these concerns in the form of a revised manuscript before we make a final decision on publication. We therefore invite you to revise and resubmit your manuscript, taking into account the points raised. Please highlight all changes in the manuscript text file.

Reviewer 1 ·

Basic reporting

The manuscript is clear and relevant, but some minors corrections should be made:

1. There is no reference to Figure 1 in the text.
2. In line 224 where it says .05 it should say 0.5.
3. In line 225 where it says .09 it should say 0.9.
4. In line 366 and 367 the authors mention that, in Figure 2, there is a 5. histogram comparing the test and retest samples. However, Figure 2 does not show this. Maybe the authors wanted to refer to Figure 4. The authors should add this figure or change their reference.
5. In line 481, it says 736. It I think that it should say 381.
6. In line 574, it says redaction it should say reduction.
7. Figure 1 says that the original sample size is 1093, but in the text, it says that it is 1035. After looking at the data in the repository, my conclusion is that the correct size of the sample is 1035 and that de the figure 1093 is a typo. Please correct this.

Additionally, I suggest some changes to improve the quality of the work:

1. Figures 2 and 3. I suggest rotating the axis 90º. With the rotated axes, there will be space in the figure for using the full name of the variables instead of their codes, improving readability. If the authors go in this direction, they could delete lines 162 to 174 from the Methods’ section.
2. The raw data is very accessible and useful. However, I would recommend, if possible, upload the scripts in raw text format and not as a pdf file which is cumbersome to use.

Experimental design

no comment

Validity of the findings

no comment

·

Basic reporting

Thank you for an interestíng, clearly written article that was supported by professional figures. It is interesting to think the whole approach from a clinical perspective (whole introduction, especially 71-75), extending it to the whole spectrum of beliefs and values.

However, I was missing references to the psychological debate about beliefs and values, especially when it comes to fringe beliefs. You, for example, mention conspiracist beliefs (63), but do not refer to theoretical and empirical works that relate conspiracist ideations to beliefs, political and otherwise (starting with Goertzl, 1994, with Swami and Uscinski as more recent contributors). I would encourage you to go into a bit more detail here.

I would also suggest to extend your discussion of Haidt, as the nature of the beliefs and values you aim to measure is crucial to assess the validity of your findings. (see 3. Validity).

You might consider to re-visit your abstract. Right now, about one third (beginning until '... in delusions') raises expectations that are not covered in your manuscript, e.g., the mechanisms of belief.

Experimental design

no comments here, procedure and descritpiton are sound and thorough

Validity of the findings

There is one aspect in your findings that has strong implications for the validity, and that is not discussed in sufficient detail yet. You refer to Haidt as a source for your morality items, and according to Haidt, judgements stemming from the proposed moral foundations should be quite stable over time. In some publications, those moral foundations are even seen as the lasting source of more variable political beliefs.

This is at odds with your findings, where the moral theme turned out to be the least stable over the course of several months. The explanation given (570-572) might or might not hold; but it should be connected to the theoretical background (Haidt/ MFT).

An alternative explanation for the rather instable dimension might be that the questions in your condensed version suffer from inherent reliability issues, as most of them focus on emotional and physical hurt. That is just one aspect of the MFT; but one aspect that might be most susceptible to a person's current affective state. As it is now, 'Hostility' would be a more fitting name for this dimension.

All other conclusions are transparent and well-stated.

Additional comments

Thank you for an interesting manuscript. I'm interested to see how your measure will perform with a clinical sample, especially with regard to the three dimensions (A/I/R). Having the full spectrum of beliefs and values in mind right from the beginning is important -- and a promising avenue.

---

## Round 0.2 · Major Revisions

· Academic Editor

Major Revisions

Your manuscript has now been re-reviewed by 2 reviewers. Although we are interested in the possibility of publishing your study, some concerns still need to be addressed before we make a final decision on publication. We therefore invite you to revise and resubmit your manuscript, taking into account the points raised. Please highlight all changes in the manuscript text file.

Reviewer 1 ·

Basic reporting

no comment

Experimental design

no comment

Validity of the findings

no comment

Additional comments

The authors addressed all my previews comments, so I recommend accepting the manuscript.

·

Basic reporting

no comment, all the points mentioned in the first round have been addressed

Experimental design

no comment, all the points mentioned in the first round have been addressed

Validity of the findings

I am still not convinced that morality, as operationalized in those studies, is on par with the other dimensions.

Beliefs and values are associated with stable predispositions. Especially, Haidt's MFT rests on the assumption of very stable underlying intuitions. When looking at the ICC confidence intervals (Koo & Li suggest this, rather than focusing on the ICC value alone), M_A with a lower bound of 0.46522 must be considered "poor to moderate"---for a rather short time interval of 48 hours. There are other hints that morality seems to be particular here (cf. line 604/ 605).

While the test-retest stability cited here by Graham et al. (2011) was not that high either, the correlation reported there was .71 for a test-retest interval of four to six weeks.

The justification given here, line 573 to 575, is not convincing, as "political or life events" that might cause attitudes to alter should not, in general, have that massive of an influence for a retest 48 hours later (for the retest 3.5 month later, that might seem more plausible). I would acknowledge that the whole politics dimension might be less stable because of current political developments reported in the news (even within 48 hours), but the items given here for morality do, prima facie, not appear to be susceptible for such short-term influence.

Additional comments

Thank you for your revised manuscript!

While everything presented here undoubtedly has great merit and should be reported, I do not share the conclusion that the BVI in its present form, with all five dimensions, is valid as an instrument that covers "beliefs and values", as there are empirical indications here that the morality dimension is on a different conceptual/ psychological level.

One possibility could be to report all the findings and propose a four-factor-BVI, while undertaking further research on the morality dimension; and/ or run a short study with just the M-items, to see if the ICC scores are comparable for a different sample (maybe it was just some unknown bias in the original sample). The "poor to moderate" ICC CI, for M_A in particular, is puzzling and should be investigated further (theoretically and empirically) before being included as a BVI dimension.

---

## Round 0.3 · accepted · Accept

· Academic Editor

Accept

We are delighted to accept your manuscript for publication.

# ·

Basic reporting

no comment

Experimental design

no comment

Validity of the findings

no comment

Additional comments

Thank you for the very interesting manuscript!

One minor comment/ suggestion, where you should check with the editorial office in case of acceptance, as it will depend heavily on the final form/ typsetting:

The grey text in Figure 2, the plots in Figure 3, the axis labels in Figure 4, and especially the numbers/ texts inside the boxes and circles of Figure 6 are hard to read on a printout, you might want to consider a final check of your Figures regarding font color and size.